# Damage Investigation in PMMA Polymer: Experimental and Phase-Field Approaches

**DOI:** 10.3390/polym16233304

**Published:** 2024-11-26

**Authors:** Lotfi Ben Said, Hamdi Hentati, Mondher Wali, Badreddine Ayadi, Muapper Alhadri

**Affiliations:** 1Department of Mechanical Engineering, College of Engineering, University of Ha’il, Ha’il City 2440, Saudi Arabia; 2Laboratory of Mechanics Modeling and Production, National Engineering School of Sfax, University of Sfax, Sfax 3038, Tunisia; 3High School of Sciences and Technology ESST of Hammam Sousse, University of Sousse, Soussse 4023, Tunisia; 4Laboratory LEE, National Engineering School of Sfax, ENIS, University of Sfax, Sfax 3038, Tunisia

**Keywords:** phase-field approach, polymer, brittle fracture, experimental analysis

## Abstract

The prediction of crack patterns is one of the main tasks in the field of fracture mechanics in order to prevent the total damage of various materials, particularly Methyl Methacrylate Polymer (PMMA). The few data in the literature underscores the need for additional experiments on PMMA to analyze the performance of the phase-field approach to predict crack trajectories. The main purpose of this study is to verify the accuracy of the phase-field approach with a staggered scheme, based on spectral decomposition, for predicting crack propagation in PMMA specimens by comparing it with the experimental results presented in this work. Based on the tensile test and SEM analysis, this material exhibits brittle behavior. The numerical approach considers cracks as diffuse damage rather than sharp discontinuities, enabling a more accurate representation of brittle fracture processes. Experimental determination of material properties is used in the development of the numerical model. The main aim of these experiments is to explore how variations in load and specific geometries influence fracture initiation and crack trajectory. Consequently, these experiments will establish a dataset to further validate numerical advancements.

## 1. Introduction

Identifying the critical load that causes crack initiation and the subsequent crack propagation patterns in polymers is crucial in engineering applications. This determination relies deeply on experimental methods but becomes particularly challenging for complex geometries. Therefore, the importance of numerical modeling is evident in predicting both the points where cracks start and the paths they follow. In this context, various damage models have been developed to predict fracture paths in both polymers and polymer composites [1,2,3]. Researchers focus on showing the accuracy of these numerical models capable of predicting crack evolution in complex structures. The primary method used for this purpose is the finite element method (FEM). This method resolves the problem of crack propagation by considering the crack surfaces as discontinuities in displacement fields. The FEM has a major difficulty when the structure presents complex crack patterns. However, the phase-field model [4,5] was developed within this framework to accurately simulate brittle fracture [6,7,8]. This model is based on Griffith’s criteria [9]. The main idea consists of introducing a scalar field to approximate the crack set. This approach offers a continuous representation of the damage field. Instead of sharp crack surfaces, it uses a diffuse interface represented by a continuous phase field. This approach, with its intricate mathematical formulations and complex numerical procedures, offers a versatile framework for simulating fracture processes in various materials and structures. Additionally, two schemes were formulated to model crack evolution using the phase-field model. The first one is the staggered scheme, widely chosen for its robustness and commonly employed in simulations [10,11,12]. Nevertheless, achieving energy convergence up to the point of brittle crack formation in brittle materials using the staggered scheme demands a high computational cost. The alternative approach, called the monolithic scheme, involves solving the displacement and damage fields concurrently [13,14,15]. This method has demonstrated to be faster compared to the staggered one [15].

The phase-field model demonstrates strong predictive capability in predicting crack propagation, particularly in complex multi-physics fracture scenarios. Within this framework, a thermo-mechanical phase-field model was formulated to forecast the evolution of multiple cracks in ceramics subjected to thermal shock [16,17]. Notably, in such instances, controlling the propagation of crack patterns experimentally proves challenging [18,19]. Furthermore, an alternative approach was elaborated to investigate ductile fracture by forecasting crack trajectories under thermal shock loading [20]. Additionally, the phase-field model found application in multifunctional materials like piezoceramics. A micro-electro-elasticity model was explained using a Ginzburg–Landau-type phase-field methodology [21]. The electromechanical coupling influences both crack patterns and the microstructural configuration, including electric polarization. In recent developments, there has been a significant advancement in phase-field models tailored to forecast crack patterns specifically in ductile materials [22,23,24]. These models incorporate the consideration of plastic dissipated energy. Additionally, researchers have explored the impact of high strain levels on crack paths and their topologies [25,26]. Furthermore, a numerical model has been devised to simulate dynamic crack propagation, yielding various crack patterns.

In the brittle fracture context, Reinoso et al. [27] conducted a study comparing experimental results with numerical predictions using the phase-field method for brittle fracture mechanics, focusing on thin composite specimens made of T700/AR-2527 material. This material exhibits brittle behavior, and the loading condition applied was Mode III fracture. The phase-field model effectively captured the crack path, showing good agreement with experimental observations. However, a slight discrepancy was noted, with the model underestimating the laminate stiffness. Xunqian et al. [28] presented a study comparing experimental results with numerical predictions using the phase-field method, applied to the brittle mortar-aggregate material. They used notched beams under bending tests, and their results showed qualitative agreement with the experimental findings of Zhang et al. [29].

In other contexts, several works present various elastic energy decomposition methods. One approach divides the strain tensor into deviatoric and volumetric components [30]. The evolution of the damage field primarily relies on deviatoric and positive volumetric parts. A spectral decomposition method can predict fissure patterns by considering crack propagation influenced by the positive eigenvalues of the elastic strain tensor [31]. Recently, a similar method was developed for the stress tensor, which is particularly useful in anisotropic elastic problems [32]. Quasi-static and dynamic phase-field models were created to compare the effectiveness of these decomposition methods, with spectral decomposition proving more suitable for brittle materials [25,33,34].

In experimental studies, few results are illustrated to prove the efficiency of the phase-field approach with the staggered scheme and based on spectral decomposition to predict crack propagation in brittle materials. The crack paths were compared between numerical results and experimental ones for the quenched C90 non-alloy steel [35]. They showed that this metal has a brittle behavior after thermal treatment. In this study, a comparison between experimental and numerical results in terms of crack growth is presented through benchmark tests. From a numerical perspective, phase-field modeling of brittle fracture has seen diverse research advancements in recent years. However, experimental validation for brittle materials remains limited. For example, Kakouris and Triantafyllou [36] simulated brittle material behavior in notched tension and shear tests, applying steel properties to the model despite steel’s inherently ductile characteristics. Similar studies have addressed the problem purely numerically, comparing phase-field schemes. The crack path predicted through spectral strain decomposition [31,35], for instance, more closely matched experimental observations than the volumetric-deviatoric strain split [30]. Other studies have also proved the effectiveness of the phase-field model in predicting brittle fracture in glass materials. For instance, Mehrmashhadi et al. [37] validated experimentally the efficiency of the phase-field model by predicting crack branching in glass induced by impact. Similarly, Schmidt et al. [38] investigated the brittle response of glass plates under bending using the phase-field fracture model. Their comparative study demonstrated that the numerical model closely matched the experimentally measured stresses and resistance of laminated glass, confirming its reliability and precision in capturing brittle fracture behavior. Additionally, dynamic crack initiation, growth, and branching phenomena in soda-lime glass were experimentally investigated by [39] using the vision-based Digital Gradient Sensing (DGS) technique combined with ultra-high-speed digital photography.

In this contribution, we expose the application of the phase-field approach for enhanced modeling of brittle fracture in PMMA. In fact, PMMA has a wide range of applications across various industries. PMMA has excellent optical clarity and transparency, making it a popular choice for various optical applications such as lenses for eyeglasses and sunglasses. It is used for automotive headlight lenses due to its high optical clarity, impact resistance, and ability to withstand UV radiation. It may be used as an alternative to glass due to its impact resistance and weatherability. In this framework, the experimental validation of the phase-field model for PMMA material has been the subject of very limited attempts. Thomas and Pollard [40] explored the two-dimensional propagation path of fractures in polymethyl methacrylate (PMMA) plates in their study. Their research delved into investigating the effectiveness of the phase-field model in predicting the quasi-static evolution of fractures in PMMA, as demonstrated by experimental results. Li et al. [41] investigated the brutal fracture phenomena in PMMA through the phase-field model, giving significant insights into the material’s response to impact loading. Their study achieved favorable outcomes, indicating the model’s effectiveness in capturing fracture patterns during impact tests. Ambati et al. [10] experimentally validated phase-field predictions using PMMA in asymmetrically notched beam and L-shaped panel tests, though only these two cases involved PMMA. Additional experiments were conducted on cement mortar, validating phase-field predictions for notched plates with a hole under tensile loading conditions.

In our study, we present experimental tests aimed at evaluating the efficacy of the proposed approach, where the developed model is validated through tensile and three-point bend experiments. The numerical simulations show a strong correlation with the experimentally observed crack trajectories and the force–displacement evolution. Additionally, several tests were conducted with variations in the shape and dimensions of the notches, and a good agreement between the numerical and experimental results was consistently observed. This paper is structured as follows: Section 2 presents the experimental setup and outcomes regarding the mechanical and fracture properties and crack patterns observed in PMMA. Section 3 offers a detailed explanation of the phase-field approach and the process of decomposing energies into tension and compression components. In Section 4, we analyze numerical results, including a comparison between experimental and predicted crack patterns from various three-point bend fixture and tensile tests, considering different specimen geometries. Additionally, we outline the experimental validation of the proposed model. Finally, Section 5 presents concluding remarks on the applicability of the phase-field approach in approximating brittle fracture in PMMA.

## 2. Materials and Methods

In this section, we outline various experimental approaches aimed at analyzing crack patterns in PMMA specimens. We focus firstly on mechanical testing of PMMA in order to define its mechanical properties. These tests were conducted at room temperature. Secondly, experimental tensile and three-point bend fixture tests will be realized on notched specimens to determine the crack pattern.

### 2.1. Materials and Experimental Setup

Standardized polymethyl methacrylate (PMMA) specimens were machined and subjected to controlled tension until failure using a universal tensile machine (HENSGGRAND—WDW-20: Jinan Hensgrand Instrument Co., Ltd., Jinan, China) (Figure 1). The thickness of these specimens was *Th =* 4 mm. Mechanical properties were obtained.

Additionally, the compact tension test for PMMA involves using a specific specimen configuration to experimentally determine fracture toughness *K_IC_*.

Applying the elastic crack tip solution, Griffith criteria [9], and Irwin’s energy theory [42], a correlation between fracture toughness, denoted *K_IC_*, and the critical elastic energy release rate, *G_c_*, was established. This relationship provides insight into how the material’s ability to resist fracture relates to the energy required for crack propagation. Then, the equation relating *G_c_* and *K_IC_* is given by:(1)$$G_{c}=\frac{{\left(K_{IC}\right)}^{2}}{E^{’}}:\;\left\{\begin{array}{ll} E^{\prime }=E & \;\mathrm{plane}\;\mathrm{stress}\\ E^{\prime }=\frac{E}{1-\nu^{2}} & \;\mathrm{plane}\;\mathrm{strain} \end{array}\right.,$$

Additionally, notched PMMA samples are tested using tensile and three-point bend fixture tests. In fact, these tests examine the crack patterns in PMMA specimens with holes and notches. By varying the distances between notches and holes, we aim to understand how these geometrical factors influence crack trajectories. The overall dimensions are held constant. In the tensile test (Figure 2a), a constant displacement *U* is applied on the bottom edge of each specimen while its top edge is fixed. In addition, the three-point bending test (Figure 2b) applies a force at the midpoint of a rectangular sample, which is freely supported at each end. The applied force is measured using a load cell, and the corresponding deflection is determined by the displacement of the system’s crosshead.

### 2.2. Mechanical and Damage Properties of PMMA

The standard PMMA sample was loaded in tension until it fractured, allowing for the measurement of various mechanical properties. Figure 3a illustrates the measured tensile curve. Tensile properties and fracture toughness data were obtained through a series of tensile tests conducted on a standard PMMA specimen and compact tension specimen. We obtain as mechanical properties Young’s modulus, *E* = 2.9 GPa; ultimate tensile strength, *UTS* = 56 MPa; elongation at break, *A%* = 2.4%; Poisson’s ratio, *ν* = 0.35; and fracture toughness, *K_IC_* = 0.8 MPa/mm^½^. A scanning electron microscopy (SEM) analysis of the fractured surface of PMMA was conducted (Figure 3b).

The curve (Figure 3a) is linear and progresses into a smooth ultimate tensile strength (*UTS*). The *UTS* is the maximum stress a material can withstand before failure. A brutal fracture occurs leading to rapid crack propagation. The SEM analysis of the fractured surface of the PMMA sample reveals predominantly quasi-brittle fracture characteristics, indicating that the material underwent limited deformation prior to failure, closely resembling brittle behavior. The observed fracture surface primarily displays features of brittle fracture, such as sharp crack paths and minimal plastic deformation, which is consistent with the quasi-static loading conditions. Furthermore, the crack propagation was predominantly quasi-static, with minimal evidence of dynamic due to speed behavior. The branching observed in Figure 3b indicates that the cracks accelerated to a certain speed, resulting in distinctive marks on the fracture surface. These branching patterns can be attributed to speed-dependent surface features, consistent with the described phenomenon.

### 2.3. Experimental Results of Various Fracture Processes in Notched PMMA Specimens

The initial experiments delve into examining the propagation trajectories of fractures within PMMA plates under three-point bending tests. Subsequently, the second experiment hones in on investigating the failure mechanisms of specimens undergoing tensile loading conditions. By conducting these experiments, a comprehensive understanding of crack patterns and failure modes in PMMA specimens can be achieved. The tests were conducted with a loading rate of 10 mm/min using a tensile machine, and the specimens were prepared through precise machining under controlled environmental conditions at a temperature of 20 °C. Each test was repeated three times, ensuring consistency in the results. For every experimental test, the same crack path was observed, demonstrating the reliability and repeatability of the testing procedure.

#### 2.3.1. Notched Specimen with a Hole Submitted to a Three-Point Bending Loading

A notched specimen with a 2 mm diameter hole was subjected to three-point bending loading, as shown in Figure 4. The specimen dimensions were 80 mm in length, 14 mm in width, and 10 mm in thickness. A support span of 60 mm was used for the three-point bending setup. The length of the bottom notch was *l* = 1 mm. The distance between the notch and the axis of the hole is denoted *c*. The distance between the axis of the hole and the bottom edge of the PMMA specimen is *d*.

Different tests were conducted, with each test repeated three times to ensure consistency in the results. For each experimental test, the same crack path was observed, demonstrating the repeatability and reliability of the testing procedure.

The experimental results, for different values of *c* and *d*, are illustrated in Table 1.

The crack pattern depends on the distance *c* and *d*. In fact, for the first test (*c* = 3.5 mm and *d* = 5 mm), the crack propagates slightly and suddenly from the tip of the notch to the top edge. However, in the subsequent test (*c* = 2 mm and *d* = 3 mm), the crack extends from the notch to the hole and after a brief time, it propagates slightly towards the top edge of the specimen.

#### 2.3.2. Double-Notched Specimen

A rectangular plate with double notches was subjected to tension (Figure 5a). The length of the specimen was 100 mm. The overall and gage widths of the specimen were 28 mm and 18 mm, respectively. The distance between the two notches is denoted *a*. The thickness of the specimens was *Th =* 4 mm. The experimental results are illustrated in the following figure.

The crack pattern depends on the distance between the two notches. In fact, for the first test (*a* = 8 mm), the crack propagates straight from the left notch to the right edge of the PMMA specimen without taking into account the existence of the second notch. However, in the subsequent test (*a* = 4 mm), the crack extends from the left notch to the right one of the double-notched specimen.

#### 2.3.3. Notched Specimen with a Hole

A rectangular plate with a straight notch and a hole with a diameter of 4 mm was subjected to tension loading (Figure 6a). The length of the specimen was 100 mm. The overall and gage widths of the specimen were 28 mm and 18 mm, respectively. The thickness of the specimens was *Th* = 4 mm. The distance between the notch and the axis of the hole is denoted *b*. The hole was in the middle of the specimen. The experimental results, for different values of *b*, are illustrated in the following figure.

The crack pattern depends on the distance between the notch and the axis of the hole. In fact, for the first test (*b* = 10 mm), the crack propagates from the left notch to the right edge of the PMMA specimen seemingly disregarding the presence of the hole. However, in the subsequent test (*b* = 4 mm), the crack extends from the left notch to the hole and after a brief period, continues to propagate towards the edge of the specimen.

## 3. Phase-Field Formulation

### 3.1. Description of Phase-Field Approach

The phase-field model, often termed the variational approach to fracture, is a numerical approach, which interests the scientific community due to the pivotal challenge in understanding fracture phenomena. In recent years, there has been significant research on phase-field models, revealing their capacity to generate intricate crack patterns by replacing sharp breaks with a scalar damage field. In fact, the variational approach in modeling brittle fracture mechanics aims to predict the initiation of cracks and their evolution through the energy minimization method. The phase-field model enables the modeling of crack propagation without prior knowledge of crack paths by introducing a continuous damage field that smooths displacement discontinuities on cracked surfaces, eliminating crack tip singularities. Consider a solid Ω with an internal discontinuity discrete crack zone Γ and an external boundary ∂Ω (Figure 7a). We approximate the damage zone Γ using a diffusive damage field denoted α by the phase-field (Figure 7b), which is zero outside the crack (α = 0) and one within the cracked region (α = 1).

In addition, during crack propagation, the degradation of stiffness, as defined in the strain energy term, was analyzed using various decomposition methods. In our approach, we utilize the spectral decomposition method to characterize this phenomenon. This method allows for distinguishing between tensile and compressive parts’ results in crack growth analysis. The decomposition of elastic energy density into positive and negative parts provides a physically meaningful framework for modeling fracture in materials. The positive part of the elastic energy density represents the energy that can be released during the initiation and propagation of cracks. Contrariwise, the negative part of the elastic energy density represents the energy that is stored within the material and opposes the initiation and growth of cracks.

### 3.2. Description of Decomposing Energies—Spectral Decomposition Method

The alternate minimization principle allows the determination of the regularized crack phase field α(x,t) in Ω space. The response of the fracturing solid is described by displacement and crack phase field.
(2)$$U:\left\{\begin{array}{c} \Omega \times T\;\to R^{3}\\ \left(x,t)\;\mapsto U(x,t\right) \end{array}\right.\;\;\;\mathrm{and}\;\;\;\alpha :\left\{\begin{array}{c} \Omega \times T\;\to \left[0,1\right]\\ \left(x,t)\;\mapsto \alpha (x,t\right) \end{array}\right.,$$

The sharp crack topology is regularized by the diffusive crack surface Γ as follows
(3)Γ=∫Ωγ α,∇αdV   with   γ α,∇α=α22η+η ∇α22,
where *ղ* is the regularization parameter. The fracture energy density is given by the following relationship:(4)Wαα,∇α=Gc γα,∇α,
where *G_c_* is Griffith’s critical elastic energy release rate. If the applied stress leads to an energy release rate exceeding *G_c_*, then crack propagation becomes energetically favorable, and fracture occurs. Given arbitrary admissible displacement and damage fields and under the small displacement theory, the elastic energy density for the variational framework is defined as follows:(5)Wel=121−α2ℂ_¯ε_U.ε_U ,
where ℂ_¯ is the isotropic elasticity tensor and ε_U is the symmetric elastic strain tensor. Taking into account the stiffness degradation derived from the elastic energy expression in Equation (5), the stress tensor is thus given by the following relationship:(6)σ=1−α2ℂ_¯ε_U ,

The potential energy density per unit volume is given by Equation (7):(7)$$W\left(\varepsilon ,\alpha ,\nabla \alpha \right)=W_{el}\left(U,\alpha \right)+W_{\alpha }\left(\alpha ,\nabla \alpha \right),$$

The potential energy, *E*(*U*,*α*), for the isotropic crack topology is given by Equation (8):(8)$$\begin{array}{ll} E(U,\alpha ) & =E_{el}(U,\alpha )+E_{\alpha }(\alpha )\\ & ={\int_{\Omega /}}_{\Gamma }W_{el}\left(\varepsilon ,\alpha \right)dx+\int_{\Gamma }W_{\alpha }\left(\alpha ,\nabla \alpha \right)d\Gamma \end{array},$$

The simulation of crack initiation and propagation is established in variational theory, aiming to identify the minimizers of the potential energy. This process involves employing a staggered scheme, which iteratively minimizes this energy. This approach simplifies the algorithm due to the reduced degrees of freedom in each subproblem. In addition, Newton–Raphson iterations is used to ensure the stability of the staggered scheme.

The growth of the crack phase field α, which represents the evolution of cracks within the material, happens only under conditions of tensile elastic energy, denoted $\Psi^{+}$. Let us designate ε^+^ and ε^−^ as the positive and negative components, respectively, of the total strain.
(9)$$\Psi^{\pm }(\varepsilon )=\frac{\lambda }{2}{\langle \mathrm{tr}\;\varepsilon \rangle }_{\pm }^{2}+\mu \;\varepsilon_{\pm }^{2}\;,$$
(10)$${\langle X\rangle }_{\pm }:=\frac{1}{2}\;\left(\;X\;\pm \;\left|X\right|\right)\;\mathrm{and}\;\varepsilon_{\pm }:=\overset{3}{\underset{i=1}{\sum }}{\langle \varepsilon_{i}\rangle }_{\pm }\;N_{i}\;⊗\;N_{i},$$
where ${\left\{\varepsilon_{i}\right\}}_{i=1}^{3}$ and ${\left\{N_{i}\right\}}_{i=1}^{3}$ are the principal strain components and directions, respectively. Based on the spectral decomposition method, the elastic energy density is decomposed on tension and compression parts. The damage degradation function affects the positive component of the elastic energy density.
(11)$$W_{el}\left(\varepsilon ,\alpha \right)={\left(1-\alpha \right)}^{2}\Psi^{+}\left(\varepsilon \right)+\Psi^{-}\left(\varepsilon \right)$$

The minimization problem of the regularized energy takes the following compact form:(12)$$(U,\alpha )=\arg\left\{\underset{U}{\min}\underset{\alpha }{\min}\left[E(U,\alpha )\right]\right\},$$

The iterative minimization task requires solving two problems. In each iteration, the first problem (U-problem) seeks to determine the strain field that minimizes the potential energy. This involves solving a linear elasticity problem with accompanying stiffness degradation. The second problem (α-problem) focuses on finding the crack phase field that minimizes the potential energy.
(13)$$(U,\alpha )=\left\{\begin{array}{l} U\text{-}\mathrm{problem}\text{:}\;\arg\left\{\underset{U}{\min}\left[E_{U}(U,\alpha ,\delta U)\right]\right\}\\ \alpha \text{-}\mathrm{problem}\text{:}\;\arg\left\{\underset{\alpha }{\min}\left[E_{\alpha }(U,\alpha ,\delta \alpha )\right]\right\} \end{array}\right.,$$

The variational formulations of the both problems are as follows:

For the displacement field (U-problem):(14)$$E_{U}(U,\alpha ,\delta U)=\int_{\Omega }\;{\left(1-\alpha \right)}^{2}\sigma^{+}:\varepsilon \left(\delta U\right)dx+\int_{\Omega }\;\sigma^{-}:\varepsilon \left(\delta U\right)dx=0,$$For the crack phase field (α-problem):(15)$$E_{\alpha }(U,\alpha ,\delta \alpha )=\int_{\Omega }\left[\left(\alpha -\left(1-\alpha \right)H^{+}\right)\delta \alpha +\left(\eta^{2}\nabla \alpha \right)\nabla \delta \alpha \right]dx=0,$$

The derived constitutive stress response is given then by equation:(16)$$\sigma (\varepsilon ,\alpha ):={\left(1-\alpha \right)}^{2}\sigma^{+}+\sigma^{-}\;\mathrm{with}\;\sigma^{\pm }=\lambda {\langle \mathrm{tr}\;\varepsilon \rangle }_{\pm }I+2\mu \;\varepsilon_{\pm }\;,$$

The crack driving force H is introduced to account for irreversibility of phase-field evolution:(17)$$H^{+}=max_{t\in \left[0,T\right]}\Psi^{+}\left(U\left(x,t\right)\right)\geq 0$$

A crack propagates when the strain energy exceeds the critical energy required for crack growth. This process is accompanied by degradation of the material stiffness. In the context of spectral decomposition, the strain energy does not contribute to crack propagation under compressive stress states. Stiffness degradation occurs throughout the brittle material, but damage evolution (α) is driven exclusively by tensile elastic energy. Spectral decomposition ensures the correct physical prediction of failure mechanisms.

## 4. Prediction of Crack Patterns in PMMA Specimens

The phase-field approach to modeling brittle fracture has gained significant attention due to its ability to handle complex crack patterns without the need for explicit crack tracking. This method is particularly useful in materials like PMMA. In our study, the phase-field model for brittle fracture of PMMA was developed within the finite element method. For the integrated displacement studies, we employed linear quadrilateral elements, specifically designated as CPE4 in the Abaqus software. To enhance the accuracy of our simulations, the mesh is deliberately refined along the anticipated crack paths. This refinement ensures that the characteristic element size (h = 0.01 mm) remains smaller than twice the phase-field length scale, 2ղ, based on the recommendations in [8,11]. The computed results demonstrate the model’s ability to replicate key characteristics of brittle fracture in PMMA seen in experimental three-point bending and tensile tests. In addition, the phase-field model, with its intrinsic length scale parameter, plays a crucial role in accurately modeling the fracture behavior of brittle materials like PMMA. By carefully selecting this parameter, we can ensure valid predictions of crack growth, fracture resistance, and process zone characteristics. Pham et al. [6] employed the phase-field model to investigate crack propagation in PMMA specimens and discussed in detail the choice of the characteristic length scale and mesh size. In their study, they modeled PMMA specimens with a fracture strength about 50 MPa. For these specimens, they selected a characteristic length scale ranging between 0.01 and 0.03, demonstrating that this range yielded accurate and reliable results. Based on their findings and the proven effectiveness of these parameters, we opted to use a characteristic length scale of 0.03 in our study to ensure consistency and accuracy in capturing the brittle fracture behavior of PMMA. For that, the length scale parameter was assumed as 2*ղ* = 0.06 mm. Furthermore, the presence of holes in the PMMA specimens is a requirement set by the designer. These holes can significantly influence crack trajectories within the specimen. This section described crack trajectories obtained through experimentation and those predicted by the phase-field model using a staggered scheme for various geometries. Comparisons between observed and predicted crack trajectories of PMMA specimens with holes revealed sensitivity to initial conditions, particularly the distance between the initial crack tip and the hole. The applied loading rate was 10 mm/min. This is only a small fraction of the Rayleigh wave speed (e.g., 0.016% of C_R_). Then, the numerical and experimental tests implied a quasi-static loading. This loading rate would not reach the high-speed dynamic fracture regime, where cracks behave like branching instability.

### 4.1. Notched Specimen with a Hole Submitted to a Three-Point Bending Loading Numerical Versus Experimental Results

The prevention of cracking in compressed areas is directly related to the evolution of the damage field *α*, which is only influenced by the tensile elastic energy. This relationship is illustrated through the analysis of the crack path in PMMA specimens undergoing loading from the following three-point bending tests. These tests consist of a supported beam that was centrally loaded. This geometry selection ensures plane strain conditions and a minimal process zone ahead of the crack tip relative to the crack length. The notched specimen, subjected to three-point bending, was constrained in the different directions at the left and right supports to fix its displacement. A vertical displacement was applied at the midpoint of the upper edge. The geometry of the notched specimen, which is submitted to a three-point bending loading, was illustrated in Figure 4. The experimental and numerical results, in terms of crack path, are illustrated in Table 2.

The evolution of force displacement is depicted in Figure 8 for notched specimens featuring a hole, which is submitted to a three-point bending loading with varying distances between the notch and the hole’s axis.

The three-point bending tests reveal that as the distance between the notch and the hole’s axis increases, the force necessary to induce failure also rises. Additionally, the numerical simulations are in exact agreement with the experimental data, highlighting a robust match between theoretical models and empirical results.

### 4.2. Notched Specimen Submitted to Tensile Loading

To predict the crack path using the proposed model, a tensile test on the asymmetrically notched specimen (Figure 5a and Figure 6a) was applied. The notched specimen, subjected to tensile bending, was constrained in the different directions along its lower edge to fix its displacement. A vertical displacement in the y-direction was applied to the upper edge of the notched specimen. The experimental and numerical results, in terms of crack path, are illustrated in Table 3.

It was observed that the crack propagation paths are influenced by defects introduced as holes and notches. The evolution of force displacement is depicted in Figure 9 for notched specimens featuring a hole, with varying distances between the notch and the hole’s axis.

The data show that the force required to cause failure increases as this distance grows. Moreover, the numerical results align precisely with the experimental findings, demonstrating a strong correlation between theoretical predictions and practical observations.

In summary, we detail the outcomes of three-point bending and tensile tests, highlighting the fracture mechanisms and crack patterns. In the validation process of the phase-field model with spectral decomposition, several numerical examples were conducted. The initial step involved verifying the model against a commonly used benchmark example to assess its accuracy and reliability. Through various simulations, the computed results obtained were compared with experimental data to evaluate the model’s performance. This validation procedure ensured that the phase-field model with spectral decomposition accurately captured the behavior of the PMMA and provided results that closely aligned with the experimental observations, thus establishing its effectiveness in simulating complex crack patterns, particularly in PMMA materials.

## 5. Conclusions

Fracturing is a complex phenomenon influenced by factors such as material properties, specimen geometry, and loading conditions. Experimental investigations provide valuable insights into the physical mechanisms behind fracturing, facilitating the development of more accurate numerical models in particular the phase-field approach for predicting crack propagation. However, the limited availability of experimental data for certain materials, such as PMMA, highlights the need for further experimental studies to validate and refine phase-field models. These experiments should focus on understanding the sensitivity of crack nucleation and propagation to variations in loading conditions, specimen geometry, and material properties. The phase-field model developed in this study showed strong predictive capabilities, accurately forecasting both crack initiation and propagation directions. The results from numerical simulations are in excellent agreement with experimental findings, confirming the reliability of the model for simulating brittle fracture.

The implications of these findings extend beyond the specific case of PMMA, as the phase-field model can be applied to a wide range of materials and fracture scenarios, offering a powerful tool for the design and analysis of fracture-resistant structures. In future research, refining the model to account for additional factors such as dynamic loading, temperature effects, and material heterogeneity could improve its accuracy and extend its applicability to more complex real-world problems.

## Figures and Tables

**Figure 1 polymers-16-03304-f001:**
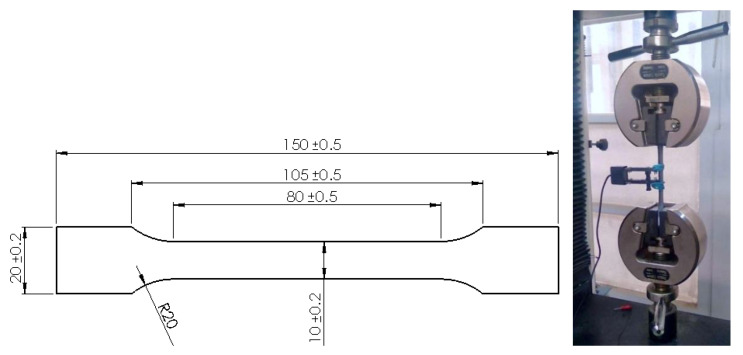
Standard PMMA sample and experimental tensile test.

**Figure 2 polymers-16-03304-f002:**
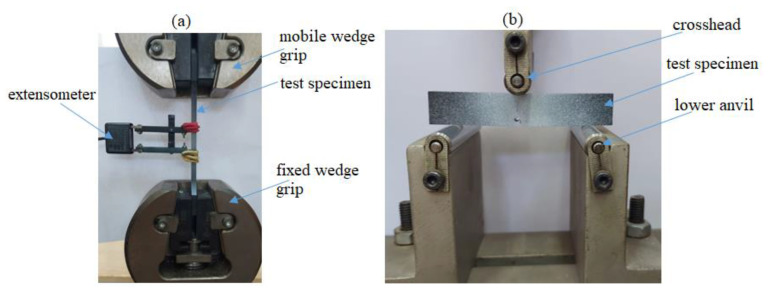
Mechanical tests for notched samples: (**a**) tensile test; (**b**) three-point bending test.

**Figure 3 polymers-16-03304-f003:**
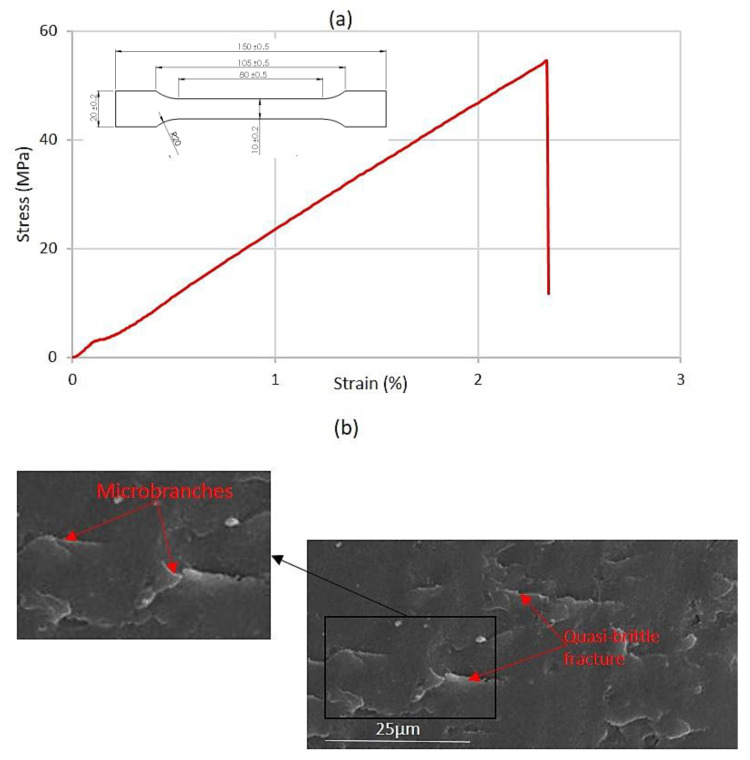
(**a**) Tensile curve of PMMA; (**b**) SEM image of fractured PMMA.

**Figure 4 polymers-16-03304-f004:**
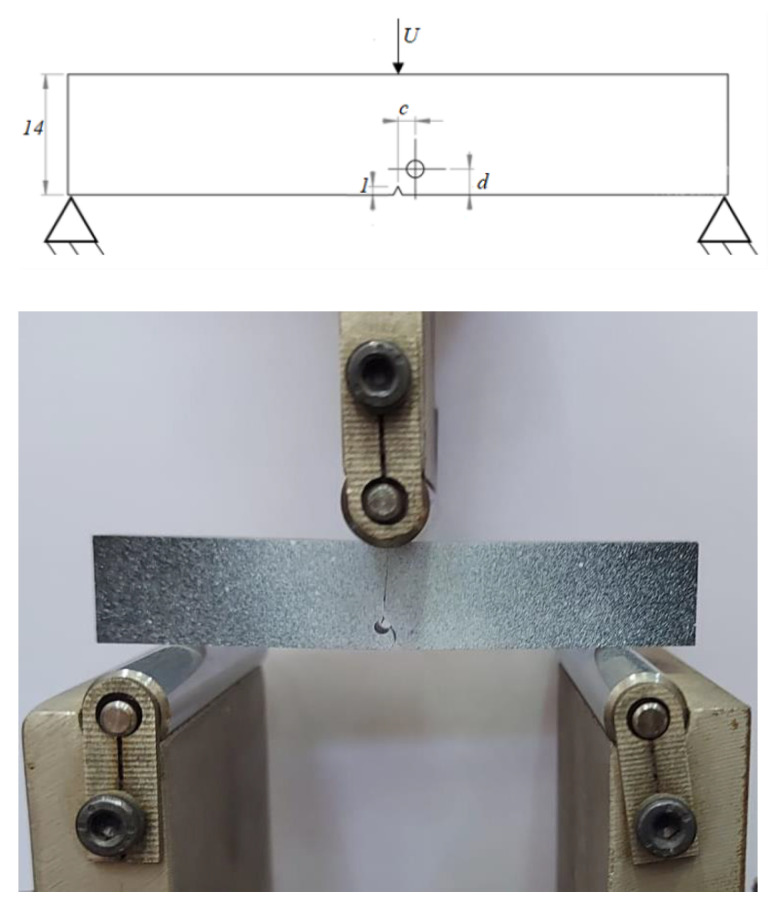
Three-point bending test of the notched specimen with a hole.

**Figure 5 polymers-16-03304-f005:**
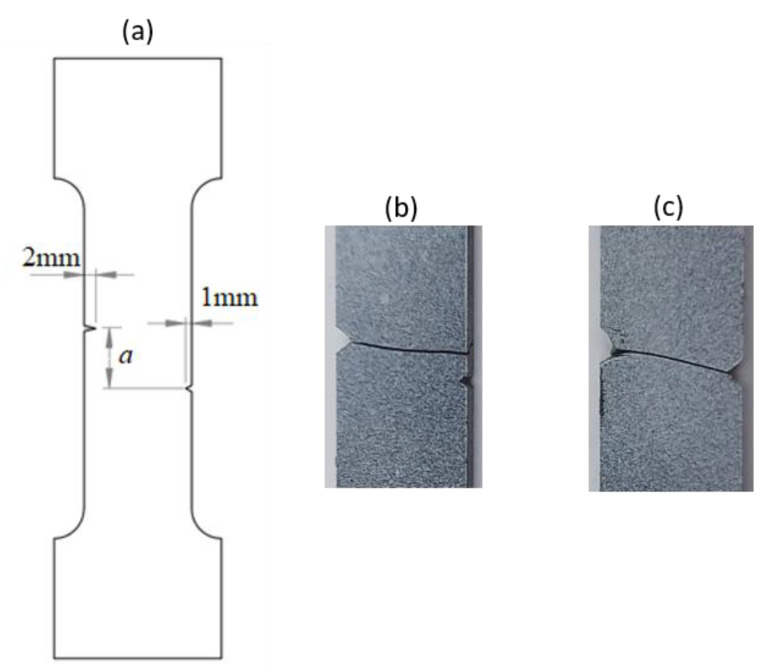
(**a**) Specimen geometry; (**b**) tensile test with *a* = 8 mm; (**c**) Tensile test with *a* = 4 mm.

**Figure 6 polymers-16-03304-f006:**
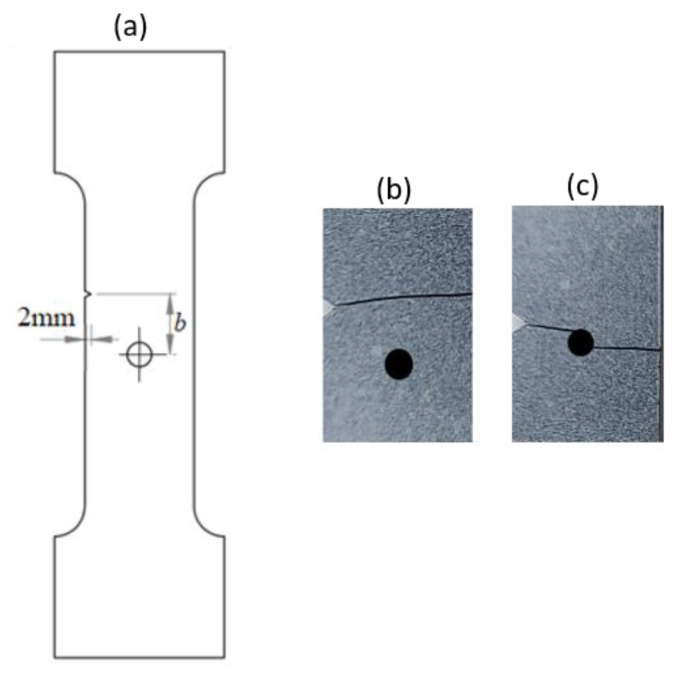
Notched specimen with a hole: (**a**) specimen geometry; (**b**) tensile test with *b* = 10 mm; (**c**) tensile test with *b* = 4 mm.

**Figure 7 polymers-16-03304-f007:**
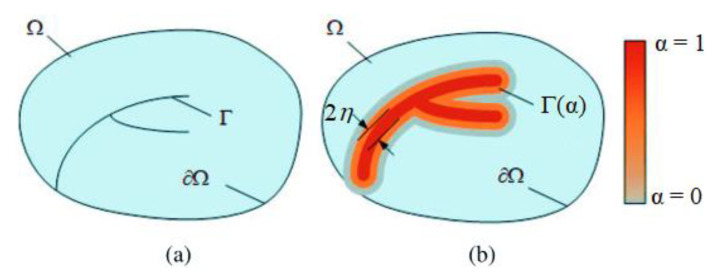
(**a**) Schematic illustration of discontinuity discrete crack zone Γ; (**b**) approximation of the diffusive damage field by the phase-field.

**Figure 8 polymers-16-03304-f008:**
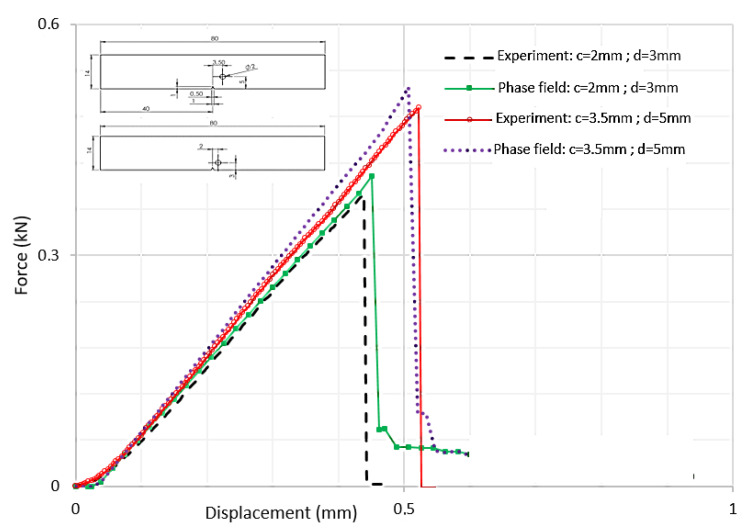
Force–displacement curves for notched specimens with a hole submitted to a three-point bending loading.

**Figure 9 polymers-16-03304-f009:**
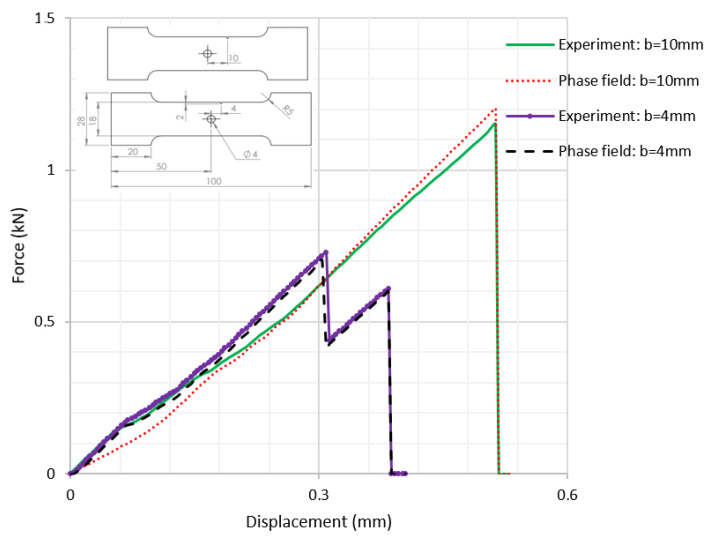
Force–displacement curves for notched specimens with a hole.

**Table 1 polymers-16-03304-t001:** Experimental results of notched specimen with a hole.

Three-Point Bending Test	Experimental Crack Paths
First three-point bending test: *c* = 3.5 mm and *d* = 5 mm	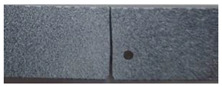
Second three-point bending test: *c* = 2 mm and *d* = 3 mm	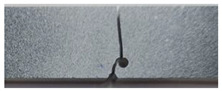

**Table 2 polymers-16-03304-t002:** Experimental and computed crack paths of notched specimens.

Tensile Test	Numerical Crack Paths	Experimental Crack Paths
First test: *c* = 3.5 mm and *d* = 5 mm	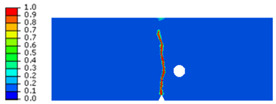	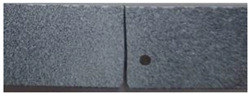
Second test: *c* = 2 mm and *d* = 3 mm	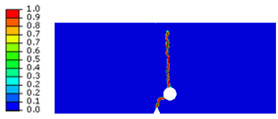	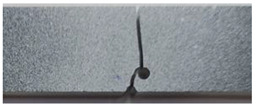

**Table 3 polymers-16-03304-t003:** Experimental and computed crack paths of double-notched specimens.

Tensile Test	Numerical Crack Paths	Experimental Crack Paths
Double-notched specimen: *a* = 8 mm	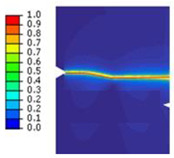	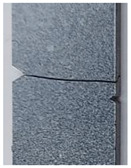
Double-notched specimen: *a* = 4 mm	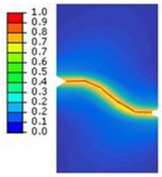	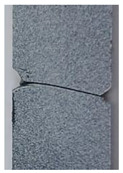
Notched specimen with a hole: *b* = 10 mm	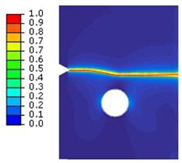	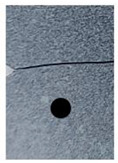
Notched specimen with a hole: *b* = 4 mm	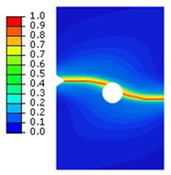	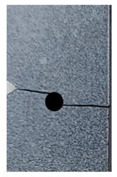

## Data Availability

The original contributions presented in the study are included in the article, further inquiries can be directed to the corresponding author.

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
