# Peer review of "Damage Investigation in PMMA Polymer: Experimental and Phase-Field Approaches"

_polymers, 2024, doi:10.3390/polym16233304_

Round 1

Reviewer 1 Report

Comments and Suggestions for Authors

The main purpose of this study is to verify the accuracy of the phase-field approach with a staggered scheme, based on spectral decomposition, for predicting crack propagation in PMMA specimens by comparing it with the experimental results presented in this work. Based on the tensile test and SEM analysis, this material exhibits brittle behavior. The numerical approach considers cracks as diffuse damage rather than sharp discontinuities, enabling a more accurate representation of brittle fracture processes. Experimental determination of material properties are used in the development of the numerical model. The work is very interesting as it deals with many industrial applications.

The topic falls within the scope of the target journal. The logic of the work has been well organized. The language is roughly smooth. The results are analyzed reasonably. It could be considered for publication after some careful modifications.

1. After each equation, there should be “,” or “.” to ensure it is a complete sentence. The following “Where” should be “where”.

2. The format “[18-19]” should be “[18, 19]”.

3. The number and its following unit should not be in the italic format.

4. Have the authors considered the damage theory based on continuum mechanics? Please refer to: Zhang et al., Enhanced CDM model accounting of stress triaxiality and Lode angle for ductile damage prediction in metal forming. International Journal of Damage Mechanics, 2021.

5. Are there any experimental verifications?

Reviewer 2 Report

Comments and Suggestions for Authors

This paper compares the experimental crack paths and those predicted by the phase-field approach, in tension and bending. However, this current work looks like a simple validation of the existing numerical approach, without sufficient novel contributions.

First, there’s not enough background search. The authors list several literatures for phase-field on ductile materials, multifunctional materials, ceramics etc., while not enough research for phase-field on brittle materials, for instance glass, are mentioned. There are enormous similar works.

Second, the paper does not seem to distinguish PMMA from idea brittle materials. As we know, PMMA is brittle (or quasi-brittle) with considerable size of process zone. In the dynamic fracture, the crack behavior is different from idea brittle materials for crack speed larger than 0.2cR. This is an essential difference between PMMA and glass. If this is not specified in the numerical approach, the phase-field model could be simply considered as those for glass, which again has enormous references. Moreover, the authors do not show how they introduce the dynamic effect of crack into the phase-field model.

The experimental setup is not well introduced. Specimen geometries, test fixtures, dimensions of these setups, and most importantly the stressing rate/strain rate is not introduced. I believe there’s something important missing. In section 2.2, the authors pull a dog bone specimen to fracture, and they present the fracture surface in Figure 1b. They claim that the fracture surface has “a consistent and uniform morphology with no notable variations or distinct patterns”. In the fracture of PMMA, the fracture surface could show different morphologies based on the crack speed traveling along the fracture surface. Based on the author’s description, I can only guess that the specimen is fractured in relatively low strength at low speed. But this is a rare case from tensile fracture without any precracks or notches. The authors should explain this better by reporting the loading rates, showing the entire fracture surface, or try to do experiments with several different loading rates, while including more details.

Reviewer 3 Report

Comments and Suggestions for Authors

Some seuggestions and questions about your work:

1. Experimental Study

  • Loading Velocity: Please specify the loading velocity (e.g., cross-head velocity or strain rate) used in the experiments to confirm that the study operates within a quasi-static framework.

  • Texture and Failure Mode Analysis: Figure 1(b) presents a textured surface that appears to support a brittle fracture under quasi-static conditions. However, recent studies suggest that PMMA may experience catastrophic failure with both quasi-static and dynamic characteristics. Did you observe any smooth regions on the fracture surface, and if so, how do these align with the dual nature of PMMA failure?

  • Test Repetition and Result Reliability:

    • How many tests were conducted, and what steps were taken to ensure repeatability of the results?
    • Have you accounted for and removed machine stiffness from your data to isolate the material response accurately?
    • Including dispersion or error bars in your results could provide valuable insights into data variability and result reliability.

2. Numerical Study

  • Phase-Field Damage Model Specification: Please specify the phase-field damage model applied in your numerical study. Additionally, elaborate on how the Kuhn-Tucker condition is verified within your model.

  • Rationale for Spectral Decomposition: What is the rationale for using spectral decomposition in this study? Providing insight into its relevance or benefits within your specific model could enhance clarity.

  • Boundary Conditions: Clearly define the boundary conditions applied in your numerical simulations to facilitate reproducibility.

  • Discrepancies Between Numerical and Experimental Results: Some numerical results differ from the experimental findings (crack path ...). Could you discuss potential reasons for these discrepancies?

  • Force-Displacement Curve Analysis: In Figure 7, the force-displacement curve produced by the phase-field model appears non-linear. Could you explain the source of this non-linearity?

Conclusion and Literature Context

  • Improving the Conclusion: Consider enhancing the conclusion to reflect a broader interpretation of your findings and their implications.

Round 2

Reviewer 2 Report

Comments and Suggestions for Authors

In the revised version of the manuscript, more references and more details about the experiments are provided. But I still have some doubts regarding the key part.

In the revised version, the authors used twice the characteristic length scale 2*eta=0.06 (what is the unit?) to specify the phase-field method for PMMA. However, using mesh size (what is the mesh size in this model?) smaller than half the characteristic length scale is also common in the phase-field of brittle material like glass. I agree with authors that at very low crack speed, the dynamic behavior of cracks in PMMA is less important, but this also dimishes the contribution of this work, making it looks like a simple verification of the basic phase-field approach in brittle fracture. If the authors think that with 2*eta=0.06 is enough for the phase-field method to work for PMMA with the process zone, please provide more details and references to convince the readers.

Regarding the dogbone test in section 2.2, it is clear that no precrack or notch is introduced. In the revised version, I now know the authors used a very low loading rate leading to a very low speed fracture of PMMA. (Actually the reported loading rate is in section 2.3 for notched specimen. Does the dogbone test in section 2.2 use the same loading rate?) However, Fig. 3a shows that the dogbone specimen breaks at 55MPa. For PMMA, 55MPa is not a very low strength, and crack can accelerate to a certain speed and leave telltale marks on the fracture surface, while the authors insist that the crack is quasi-static and there's no special speed-related surface features, which is against my experience. I suggest again that the authors show the entire fracture surface, and they can comment on the features of quasi-brittle fracture in a zoom-in subfigure like the current Fig. 3b.

Furthermore, if authors add descriptions like "fracture surface primarily exhibited characteristics of brittle fracture, including sharp crack paths and minimal plastic deformation", they should also modify the original observation "uniform morphology with no notable variations or distinct patterns".

I think the authors should also include some references in the introduction regarding the classic work of phase-field on brittle fracture, e.g., glasses.

In the legend of Fig. 9, b=5mm, while in Table 3, b=4mm.

Reviewer 3 Report

Comments and Suggestions for Authors

All comments and corrections were taken into account.

Author Response

We appreciate all of your insightful comments. We thank you kindly for taking time and energy to provide such constructive comments on the proposed work and to help us to improve the paper.

Round 3

Reviewer 2 Report

Comments and Suggestions for Authors

Please add the decription of how to select proper characteristic length scale (and its reference) in Section 4.

In Section 4, h=0.01mm (also need the unit).

Regarding the dogbone test, what I suggested is to provide an overall image of the fracture surface. In the attached file I provide an example. This is a PMMA brick fractured at 29MPa, much lower strength than your dogbone test. On its fracture surface, different regions including static growth region of the inital flaw, speed-related directional conic marks, and some rougher regions with higher speed, could be clearly observed. Your specimen was fractured at higher strength hence it is supposed to have higher crack speed than the attached example and is supposed to show more telltale surface marks. If your fracture surface is not similar to the example, which is okay, just provide the overall view of the fracture surface for the reader's reference. With updated figures, the description in Section 2.2, page 6, should also be modified. The descriptions should correspond to what readers can see in the figure.

Also, I suggest to use "microcracks" or "microbranches" instead of "branches" in Fig. 3b. Branching usually refers to crack bifurcation at very high speed, while microbranching is related to local crack speed and the mircostructure of the material.
